# *Decreased Spikelets 4* Encoding a Novel Tetratricopeptide Repeat Domain-Containing Protein Is Involved in DNA Repair and Spikelet Number Determination in Rice

**DOI:** 10.3390/genes10030214

**Published:** 2019-03-13

**Authors:** Shen Ni, Zongzhu Li, Jiancheng Ying, Jian Zhang, Hongqi Chen

**Affiliations:** 1State Key Lab of Rice Biology, China National Rice Research Institute, Hangzhou 311400, China; asen1225@126.com (S.N.); yingjiancheng1@126.com (J.Y.); 2Department of Biochemistry and Molecular Medicine, The George Washington University, Washington, DC 20037, USA; lizzh07@hotmail.com

**Keywords:** rice (*Oryza sativa* L.), DNA repair, spikelet number, tetratricopeptide, LRR (leucine-rich repeat)

## Abstract

Spikelet number per panicle is a determinative factor of rice yield. DNA repair epigenetically alters the DNA accessibility, which can eventually regulate the transcription of the target genes. However, what and how DNA repair genes are related to rice spikelet development remains unknown. Here, we report the map-based cloning of a novel spikelet number gene *DES4* encoding a tetratricopeptide domain-containing protein. DES4 is a close ortholog of *Arabidopsis* BRU1, which is functionally related to axillary meristem development. A single base pair deletion in the last exon of *DES4* caused a premature stop of the resulting protein. The *des4* mutant exhibited dwarf, reduced tiller, and spikelet numbers phenotypes, as well as hypersensitivity to genotoxic stresses, suggesting its essential role in DNA repair. *DES4* is predominantly expressed in young panicles and axillary meristems, and DES4 protein is localized in nucleus. A set of DNA repair genes such as *cyclins*, *KUs* (*KD subunits*) and *recombinases* were differentially regulated in *des4*. Meanwhile, rice spikelet number genes *LAX1*, *LAX2*, and *MOC1* were significantly down-regulated in *des4*. In morphology, *des4* showed more severe reduction of spikelet numbers than *lax1*, *lax2*, and *moc1*, suggesting that *DES4* may work upstream of the three genes.

## 1. Introduction

Rice (*Oryza sativa* L.) is the major source of calories for over half of the world’s population. Meanwhile, rice is also used as a model species for plant molecular biology research due to its completed genome sequencing, mature genetic transformation technology, and genome co-linearity with other crops [1]. Rice yield is a complex agronomic trait determined by three main component traits: number of panicles per plant, number of grains per panicle, and grain weight [2]. Based on morphological dynamic changes, Ikeda et al. (2014) tentatively divided rice panicle development into nine successive stages, spanning from establishment of rachis meristem (RM) to mature panicle [3]. At the onset of panicle differentiation, RM initiates from the shoot apex meristem (SAM), and subsequently produces a number of lateral meristems, which can develop into primary branches. Similarly, second or higher order branches may arise from lateral meristems on each of the primary branches, and spikelet meristems on the secondary branches finally develop into spikelets [2,3,4,5]. In general, spikelet numbers per panicle are mainly determined by the number of primary and secondary branches on the panicle [2,6]. Numerous spikelet number genes and quantitative trait loci have been identified and extensively reviewed in several publications [2,3,5,6,7]. These include *Frizzle Panicle* (*FZP*), *LAX Panicle 1* (*LAX1*), *LAX2*, *Small Panicle* (*SPA*), and *Aberrant Panicle Organization 1* (*APO1*), which are implicated in the initiation of RM or conversion of RM into spikelet meristem (SM) [4,8,9,10]; and *Grain Number 1a* (*Gn1a*), *Dense and Erect Panicle 1* (*DEP1*), *DEP2*, and *DEP3*, which are related to the rate of spikelet differentiation [11,12,13,14]. Regardless of the progress achieved, our understanding of the regulation of spikelet number in rice remains poor, largely due to the fragmented information from a limited number of identified genes. Here, we report the positional cloning of a novel rice spikelet number gene *Decreased Spikelet 4* (*DES4*), which is functionally involved in both DNA damage repair and spikelet number determination. DES4 is a nuclear-localized, tetratricopeptide repeat domain-containing protein similar to BRU1 in *Arabidopsis*. A set of DNA damage repair genes and spikelet number genes are differentially regulated in the *des4* mutant. This work represents a substantial advance toward the understanding of how DNA repair genes are involved in rice reproductive tissue differentiation and development.

## 2. Materials and Methods

### 2.1. Plant Materials and Growth Conditions

The recessive mutant *des4* was isolated from a γ irradiation-induced mutant population in the background of Zhonghua 11 (ZH11) (*O. sativa*, ssp. japonica). Genetic mapping population was derived from a cross between *des4* and 9311 (*O. sativa*, ssp. indica). From 2014 to 2018, plants were cultivated in the experimental field of China national rice research institute (May to October) or in Sanya, China (November to March of the following year). Under natural growing conditions, morphological analysis was done on plants in the harvesting stage with fully matured grains. 

### 2.2. Map-Based Cloning

Simple sequence repeat (SSR) and sequence-tagged site markers were derived from (http://www.gramene.org/microsat/ssr.html; Appendix A), and screened for polymorphisms in the population. A DNA pool of twelve *des4* mutants were used for the draft mapping of the *DES4* locus by screening over 120 polymorphic SSR markers. To fine map *DES4*, about 1002 recessive individuals with the mutant phenotype were selected and screened by newly developed SSR and Indel markers. Primer sequences of the polymorphous markers for fine mapping can be found in Appendix A.

### 2.3. Plasmid Construction and Plant Transformation

For complementation of the *des4* mutant, full coding sequence (CDS) of the candidate gene was amplified from the ZH11 panicle cDNA and cloned into expression vector pCambia1300-221-myc, in which the gene is fused with myc peptide tag sequence and driven by a 35S constitutive promoter. The plasmid was then introduced into *des4* embryonic calli by *Agrobacterium tumefaciens*-mediated transformation according to a previous report [15]. Similarly, full CDS of *DES4* was cloned into pCambia1300-YFP using the SalI and SmaI sites to fuse with the yellow fluorescent protein (YFP) ORF on the N terminus. pCambia1300-YFP-DES4 and the negative control pCambia1300-YFP plasmids were transiently transformed into tobacco leaf epidermal cells as previously described [16]. Fluorescence was observed on confocal microscopy (Zeiss LSM710, Carl Zeiss, Jena, Germany) at 72 h after infiltration.

### 2.4. RNA Preparation, RT-PCR Analysis, and mRNA in-Situ Hybridization

Total RNA from various tissues was isolated by using TRIeasyTM Total RNA Extraction Reagent (Yeasen, Shanghai, China). First strand cDNA was transcribed from DNase I-treated RNA using M-MLV reverse transcriptase and oligo (dT) primers (Takara, Dalian, China). qRT-PCR was carried out on a CFX96 touch real-time PCR detection system (Bio-Rad, Hercules, CA, USA) by following a previous publication [16]. Actin gene *LOC_Os03g61970* was used as an internal control. Expression was assessed by evaluating threshold cycle (CT) values and calculated by the 2^−ΔΔCT^ method. The experiment was performed in biological triplicates. The mRNA in situ hybridization was conducted as described by Zhang et al. (2010) [17]. Primer sequences are listed in Appendix A.

### 2.5. Genotoxic Stress

One DAG (day after germination) seeds were hydroponically cultured on Hoagland solution with the addition of methyl methanesulfonate (MMS) and zeocin in different concentrations, in a growth chamber (28 °C, 60% humidity, 12 h light/12 h dark). The Hoagland solution was changed every three days to maintain a stable solution pH. The plant heights were manually measured after two weeks of growth. Twenty biological sample repeats were measured for each treatment.

## 3. Results and Discussion

### 3.1. des4 Has Reduced Spikelet Number

In an effort to clone rice spikelet number genes, our lab identified a set of *des* mutants (designated as *des1* to *des5*) by screening a γ irradiation-induced mutant population in the background of ZH11 (*O. sativa* ssp. japonica). Among the mutants, *des4* displayed pleiotropic phenotypes, including dwarf, less tillers, and narrower leaf blades (Figure 1A,C,D). More importantly, the primary and secondary branch numbers of *des4* were severely decreased when compared with wild type (WT), which finally led to a significant reduction of spikelet numbers per panicle from 125.4 ± 19.91 (WT) to 38.4 ± 10.38 (*des4*) (n = 15, *p* < 0.01) (Figure 1B,D).

### 3.2. Positional Cloning of des4

To clone the gene responsible for *des4* phenotypes, we adopted a map-based cloning strategy by using an F_2_ population derived from the cross between *des4* and 9311 (*O. sativa* spp. indica). We phenotypically characterized 92 F_2_ lines; the normal and *des4* plants followed a 3:1 ratio, suggesting that the *des4* phenotype is controlled by a single recessive gene (normal: *des4* = 68:24, χ^2^ = 0.0579 < χ^2^_0.05,1_ = 3.84). Approximately 4000 individual F_2_ plants, including 1002 *des4* plants, were used for the genetic mapping. We gradually mapped *des4* to a 100 kb region between marker M137 and M128 on the long arm of chromosome 2 (Figure 2A). As shown in the rice annotation project database (RAP, https://rapdb.dna.affrc.go.jp/), a total of 18 candidate genes, including 2 transposon genes were located in this region. Subsequently, we Sanger sequenced the 16 non-transposon genes in both WT and *des4*. None of the candidate genes were mutated, except a cytimidine deletion that occurred on the 3-prime ends of the CDS of *Os02g0782800*, which should have shifted the open reading frame and caused premature stop of the resulting protein (Figure 2B, Appendix A). To the best of our knowledge, no publications regarding the functions of *Os02g0782800* are available.

In terms of structure, *Os02g0782800* is around 16 kb in length with 20 exons and 19 introns, and the encoded protein is comprised of 1338 amino acids (Figure 2B). Due to the huge size of *Os02g0782800*, we failed to make a complementation vector carrying the whole genomic sequence of the gene with its native promoter (over 18 kb). Alternatively, we constructed the full CDS of *Os02g0782800* into pCambia1300-221-myc, in which the gene is fused with myc-peptide tag sequence under the driving of 35S constitutive promoter. The constructs were genetically transformed into the *des4* background. As expected, all the T-DNA positive plants (n = 4) were rescued as WT. Thus, we concluded that *Os02g0782800* is *DES4* (Figure 2C, Appendix A).

### 3.3. DES4 Encodes a Novel Tetratricopeptide Repeat Domain-Containing Protein

Conserved domain search on the DES4 sequence identified seven tetratricopeptide repeat (TPR) domains on the N terminus, eight leucine-rich repeat (LRR) domains on the C terminus, and two nuclear localization signal (NLS) sequences in the middle (Figure 3A). As a commonly identified motif in various proteins, TPR is usually comprised of 3–16 tandem repeats of a degenerate 34 amino acid sequence motif unit, which may serve as scaffolds for protein–protein interactions and protein complex assembly. TPR domain-containing proteins have been found to be extensively involved in cell cycling, gene transcription regulation, and protein transportation, as well as protein folding [18,19]. Meanwhile, LRRs are composed of repeating 20–30 amino acid stretches that are rich in hydrophobic leucines. Similar to TPRs, LRRs are also functionally related to the formation of protein–protein complexes [20,21]. Using the DES4 protein sequence as a query, we identified DES4 orthologs in 19 other plant species with a single copy in each one. Phylogenetic analysis revealed that DES4 exhibited the highest similarities with orthologs in *Brachypodium distachyon*, *Triticum urartu* and *Aegilops tauschii*, while all the members in grass were clustered in a major clade, suggesting that those proteins may have conserved functions (Figure 3B). In *Arabidopsis thaliana*, DES4 ortholog AT3G18730 has been reported as BRU1/Tonsoku/Mgoun3 participating in meristem cellular organization and maintenance. In *bru1*, the plant became dwarf with distorted phyllotaxy, brush-like branches, and defected inflorescence architectures, which mimicked the phenotypes observed on *des4*, suggesting that *DES4* may have conserved functions as *BRU1* [22,23,24]. 

We examined the mRNA level of *DES4* in various tissues, including root, leaf blade, internodes, and young panicle. *DES4* was constitutively transcribed in all the tested tissues, while the highest levels were detected in axillary buds and young panicles (Appendix A). Employing mRNA in situ hybridization technique, we further gained an in-depth view of *DES4* expression pattern in differentiating tillers and panicles. Strong signals were detected on the lateral meristems where tiller buds or panicle primary branches initiated, which is in agreement with the decreased tillers and spikelet numbers observed in *des4* (Figure 3C,E). Moreover, the NLS on DES4 invited further investigation of its subcellular localization. As shown in Figure 3F, transiently expressed DES4-YFP proteins were detected in the nucleus of tobacco leaf epidermal cells, whereas the empty vector control showed fluorescence in various cellular compartments as expected. This observation is also consistent with the subcellular localization of BRU1 in *Arabidopsis*. 

### 3.4. des4 Is Hypersensitive to Genotoxic Stress

Previous study indicated that BRU1 is crucial for the recognition and repair of DNA damage [24]. To test the potential role of *DES4* in DNA repair, we treated young seedlings of *des4* and WT with MMS and zeocin in a series of concentrations. MMS and zeocin have been found to alkylate DNA, which mimics DNA double strand break (DSB) damages [25]. As MMS concentration increased, WT seedlings displayed retarded growth (Figure 4A,B). Interestingly, *des4* seedlings were hypersensitive to the MMS-induced genotoxic stress. When <15 ppm MMS treatment was applied, the height of *des4* plants was only half that of the WT. We obtained a very similar result from the zeocin treatment (Figure 4B). Therefore, we suggest that DES4 may play an essential role in the DNA repair of DSB damage. DNA repair has been linked with transcriptional gene regulation, as both processes are essentially involved in chromatin compaction and DNA accessibility under very common mechanisms [26]. A few examples are transcription factor IIH (TFIIH) which is required for initiation of RNA transcription, efficient repair of DNA by nucleotide excision in yeast [27]; and BRU1, which has dual roles in DNA damage repair and epigenetic gene silencing in *Arabidopsis* [24]. Given the high homologies between BRU1 and DES4, as well as similar genotoxic hypersensitivities observed in both mutants, we speculate that DES4 may also have conserved functions similar to BRU1 in gene transcriptional regulation.

### 3.5. DSB Repair Genes and Spikelet Number Genes Are Differentially Regulated in des4

To investigate the potential downstream genes regulated by *DES4*, we performed qRT-PCR experiments using 14-day-old seedling cDNAs of WT and *des4* as templates. The candidate genes included seven cyclin genes, two KU homolog genes, and six recombinase genes which are potentially involved in DSB repair (Figure 5). Severe DNA damage could directly influence cell cycle and proliferation. Culligan et al. (2004) found that *Arabidopsis* seedling cells were blocked outside of M phase at G2 and/or at G1/S transition when plants were irradiated with a high dosage [28]. In response to DNA damage, some key cell-cycle-related genes such as *CDKB1;2*, *CDKB2;1*, and *CyclinB1;2* were down-regulated, whereas *CyclinB1;1* was up-regulated in the first few hours and then started to decline after 24 h of treatment [29,30]. In *des4*, we also found that the transcription of four cyclin genes (*CyclinA1;1*, *CyclinA1;K*, *CyclinA3;1*, and *CyclinB2;K)* were significantly reduced, while *CyclinD3;1* and *CyclinD4;1* exhibited higher expression level when compared with the WT. To repair the DSB, eukaryotes developed a nonhomologous end joining (NHEJ) mechanism, which is heavily modulated by DNA-dependent protein kinase complex (DNA-PKcs) containing *Ku70/Ku80* heterodimer and DNA ligase4 [31]. Ku70/Ku80 recognizes and binds to the DSB ends, while ligase4 is responsible for the repair of DNA breaks [32]. In our case, *des4* showed elevated levels of *Ku70* and *Ku80*. The suggestion is that DNA-PKcs may work in parallel with *DES4* as an alternative pathway to compensate the loss of function of *DES4*, when DNA repair is required. Meanwhile, a list of recombinase genes including *RAD50*, *RAD51A2*, and *RAD51C* were significantly up-regulated in *des4*. *RAD51s* have been known as essential proteins in homologous recombination and recombinational repair of DSB. Both *RAD51A1* and *RAD51A2* possess homologous pairing activity, though *RAD51A2* has much higher activities than *RAD51A1* [33]. *RAD51C* was also implicated in DNA repair in somatic cells [34]. The up-regulation of the RAD51s in *des4* imply that, in addition to the NHEJ, homologous repair may also take place to compensate the impaired function of *DES4*. Interestingly, we found that the transcription of *RAD17* was down-regulated in *des4*, possibly as a compensation for these up-regulated *RAD* sibling genes. It should be noted that the altered expression of these DSB repair-related genes may be an indirect consequence of the loss of function of *DES4*, and their relationship with *DES4* needs to be further explored with more direct evidence, such as multiple gene mutations and genetic complementation. Given the decreased spikelet numbers of *des4*, we also examined the transcriptional levels of three reported spikelet number genes—*LAX1*, *LAX2*, and *MOC1*—in young panicles. The results showed that all three genes were dramatically decreased in the mutant. We compared the extent of spikelet number decreases in *lax1*, *lax2*, *moc1*, and *des4*. It seems that *DES4* is an up-stream regulator of the other three genes, because it showed the strongest phenotype (Appendix A). As discussed above, the TPR domain-containing protein DES4 may regulate target gene transcription by epigenetically modulating the chromatin configuration and DNA accessibility. We believe that more spikelet number genes are subject to the regulation of *DES4*.

## Figures and Tables

**Figure 1 genes-10-00214-f001:**
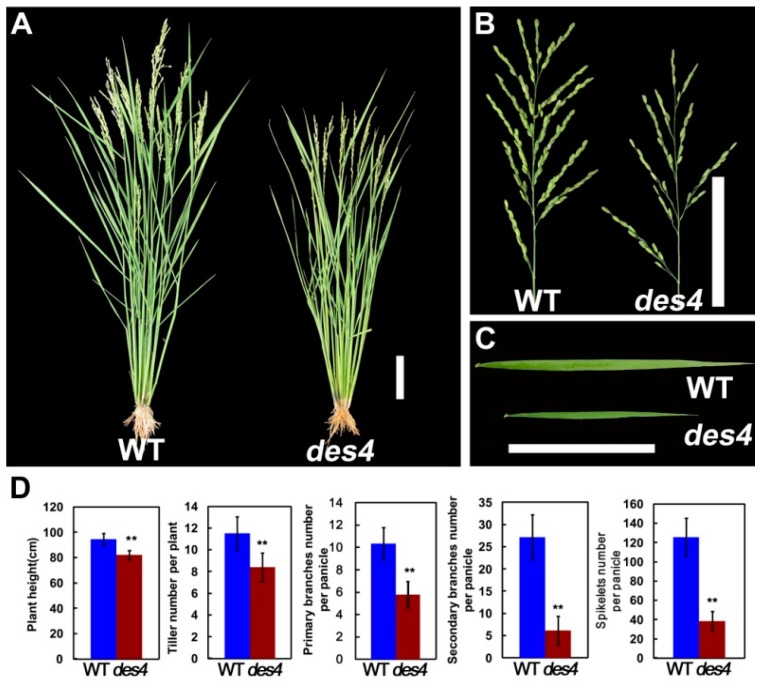
Phenotypical characterization of the *des4* and wild type (WT). (**A**) Plant architecture, (**B**) panicle structure, and (**C**) leaf shape of *des4* and WT, bar = 10 cm. (**D**) Quantification of the plant height, tiller number, primary branches, secondary branches, and spikelet numbers per panicle. Numerical values are expressed as the mean; error bars denote one standard deviation of the mean; asterisk denotes significant difference between mutant and WT within each treatment (** *p* < 0.01; Student’s *t*-test).

**Figure 2 genes-10-00214-f002:**
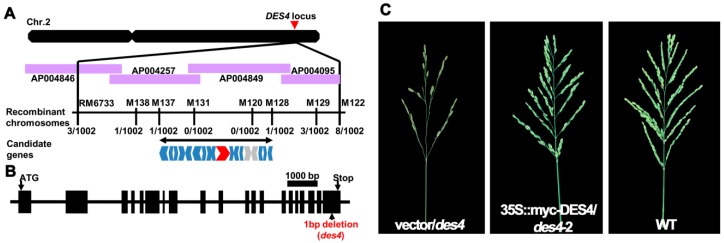
Genetic mapping and complementation analysis of *DES4*. (**A**) Map-based cloning of *DES4*. The markers for PCR-based mapping are listed in Appendix A. “Recombinant” indicates recombination between maker and phenotype; the numbers indicate the numbers of recombinant lines out of 1002 mutant type lines analyzed. Red shape indicates *Os02g0782800* candidate gene; blue shapes indicate the 16 sequenced candidate genes; and grey shape indicates the transposon genes. (**B**) Schematic presentation of the gene structure of *DES4*. Boxes represent the exons and the line indicates the intron. One base pair deletion occurred on the last exon of *DES4*. The mutated sequence is presented in Appendix A. (**C**) Panicle architectures of negative control (empty vector) line, genetically complemented line, and WT.

**Figure 3 genes-10-00214-f003:**
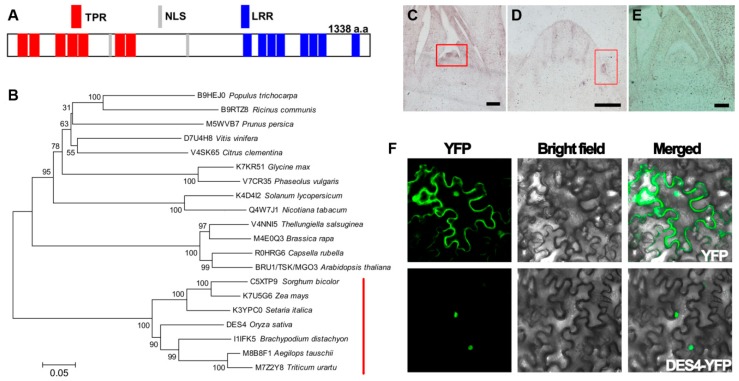
Protein structure, phylogenetic, and expression analysis of DES4. (**A**) Conserved domain analysis of the DES4 amino acid sequence. (**B**) Phylogenetic tree of DES4 and orthologs from other species. Numbers are the bootstrap values. Red line indicates the clade of grasses. (**C**–**E**) mRNA in situ hybridization analysis of *DES4* in differentiating panicles meristems (**C**) and tiller meristems (**D**). Boxes highlight the tissue parts with strong expression signals. Bar = 10 μm. (**E**) Negative control using sense probes. (**F**) Subcellular localization of DES4 in tobacco leaf epidermal cells.

**Figure 4 genes-10-00214-f004:**
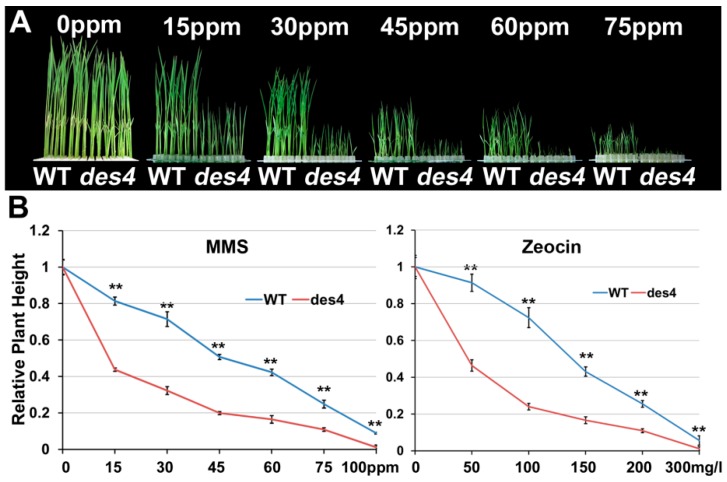
Genotoxic stresses on des4 and WT seedlings. (**A**) Growth of *des4* and WT seedlings under different methyl methanesulfonate (MMS) concentrations. (**B**) Quantification of the seedling heights under different concentrations of MMS and zeocin. ** *p* < 0.01.

**Figure 5 genes-10-00214-f005:**
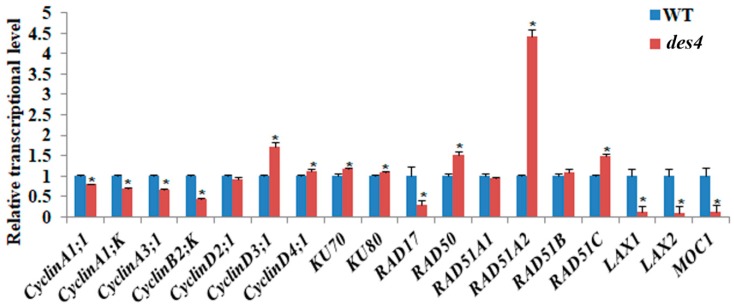
qRT-PCR analysis of double-strand break (DSB) repair genes and spikelet number genes. qRT-PCR analysis of transcript levels of DSB repair genes and spikelet number genes in WT and *des4*. Rice actin (*LOC_Os03g61970*) was used as an internal control. The locus ID for each gene can be found in Appendix A. Values are means ± SE. Asterisks indicate significant differences between WT and *des4* (* *p* < 0.05; Student’s *t*-test).

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
