# Peer review of "Decreased Spikelets 4 Encoding a Novel Tetratricopeptide Repeat Domain-Containing Protein Is Involved in DNA Repair and Spikelet Number Determination in Rice"

_genes, 2019, doi:10.3390/genes10030214_

Round 1

Reviewer 1 Report

In this paper, a gene involved in DNA repairing and spikelet number determination was isolated by map-based cloning. The des4 plant induced by gamma irradiation showed significant difference in agronomic traits such as spikelet number per panicle. DES4 was mapped on the long arm of chromosme 2. Sequence analysis revealed that a single deletion was occurred on the 20th exon of Os02g0782800. Complementation analysis showed that Os02g0782800 is DES4. The des4 mutant was hypersensitive to genotoxic corresponding to the expression patterns involved in DSB repairing. Moreover, the expression of spikelet number genes were also downregulated. Overall, this work provides a novel finding with sufficient set of evidence that DES4 is involved in DNA repairing and spikelet number determination. However, the given manuscript needs to be improved in few parts.

1. Major corrections

1) Please provide the expression data of des4 in various rice tissues.

2) Please follow the instruction that the journal has provided for dividing manuscript section. It is recommended to divide the “Results and discussions” into two separate sections, “Results” and “Discussion”.

2. Minor corrections

1) L20 : Since you have not shown the difference of the protein structure between ZH11 and des4 protein, It is inadequate to conclude that a single base deletion in the exon of DES4 caused premature protein.

2) L48 : SM → spikelet meristem(SM)

3) L90 : actin gene → Actin gene

4) L96 : MMS → methyl methanesulfonate(MMS)

5) Figure 1 : Which stage/part of the plant/sample was used for comparing phenotypic characters between WT and des4.

Author Response

Response to reviewer 1:

Please provide the expression data of des4 in various rice tissues.

Response: as we indicated by the end of the sentence, the data was actually provided as Figure S2.

2) Please follow the instruction that the journal has provided for dividing manuscript section. It is recommended to divide the “Results and discussions” into two separate sections, “Results” and “Discussion”.

Response: it is routine to present separated “Results” and “Discussion”, or combined both together as one section. Because we are submitting this MS as a short communication with very concise discussions, we decided to combine the both parts for a better layout of the text.

2. Minor corrections

1) L20 : Since you have not shown the difference of the protein structure between ZH11 and des4 protein, It is inadequate to conclude that a single base deletion in the exon of DES4 caused premature protein.

Response: Based on the CDS sequence, we deduced the premature stop of the resulting protein; evidences were actually provided in supplemental data 1.

2) L48 : SM → spikelet meristem(SM); 3) L90 : actin gene → Actin gene; 4) L96 : MMS → methyl methanesulfonate(MMS);

Response: Corrected.

5) Figure 1 : Which stage/part of the plant/sample was used for comparing phenotypic characters between WT and des4.

Response: morphological analysis was done on plants in harvesting stage with fully matured grains. Details have been included in section 2.1. The authors greatly appreciate your time and efforts for this MS.

Reviewer 2 Report

Manuscript ID: genes-445709

Title: Decreased spikelets 4 encoding a novel tetratricopeptide repeat domain-containing protein is involved in DNA repairing and spikelet number determination in rice

Authors: Shen Ni, Zongzhu Li, Jiancheng Ying, Jian Zhang *, Hongqi Chen *

Submitted to section: Plant Genetics and Genomics.

This manuscript describes the characterisation of a rice mutant affected in yield and more particularly in spikelets number per panicle (the rice inflorescence). The mutant decrease spikelet4 (des4) was isolated from a gamma irradiation-induced mutant population. The phenotype analysis indicated that the mutant is affected in various aspects of plant development (plant height, tiller number, leaf size, panicle number, panicle architecture and spikelet number). Map-based cloning and complementation assay led to the identification of a gene encoding a LRR and tetratricopeptide repeat domain-containing protein orthologous to the MGOUN3 protein from Arabidopsis thaliana. The DES4 gene is expressed in various tissues/organs tested, in axillary meristems at vegetative and reproductive phases. The authors have shown a hyper-sensitivity of genotoxic stresses of the mutant and alternation of expression of genes related to cell cycle, DNA repairing and plant/panicle architecture. As a conclusion, the author stated that DES4 is involved in DNA repairing and spikelet number determination.

General comments:

This work is interesting and is the first report on MGOUN3-like gene characterisation in rice. A lot of experiments were conducted to characterise the mutant and the function of this gene. However I think that the title and the conclusion are going too far according to the data presented in this manuscript. There is no clear evidence from the data that DES4 is involved in DNA repairing. Moreover DES4 can’t be considered as a spikelet number determination gene stricto sensu but rather a gene that may affect both vegetative and reproductive meristem functioning. In this sense, it would have been important to get a histological description of the meristems in des4 mutant in order to evidence a putative mis-organisation as reported in A. thaliana, as well as phenotypic analysis of spikelet, culm and root development in des4 mutant. Panicle images from complementation assay plants and panicle branching mutants (Fig. 2 and Fig. S3) were shown but it would be nice to get scoring (in supplementary material?) as for the des4 mutant panicles (Fig. 1). The results and discussion section related to the qRT-PCR data of marker genes is too speculative and in some cases the differential of expression between des4 and the WT is too small to be able to get clear conclusion. The authors should be more cautious about their conclusions.

Specific comments:

.    Line 20. Change “premature of » by « premature stop of »

.    Line 21. Change “The des4 exhibited” by ‘The des4 mutant exhibited”

.    Line 24. Change « nuclear » by « nucleus ». Change « serial » by « set »

.    Lines 25-27. This sentence is too long not clear.

.    Line 32. Oryza sativa in italic

.    Line 37. Delete « (also called inflorescence) ». 
 Panicle is one of the inflorescence types.

.    Line 48. Define SM (not mention before).

.    Line 50. Change « progress has been achieved » to « progress achieved »

.    Line 55. Change « serial » to « set »

.    Line 58. Change « involve » to « are involved ».

.    Lines 64-65. Which years ?

.    Line 79. DES4 in italic

.    Line 90. Which Actin gene (LOC or RAPDB locus name) was used as qRT-PCR reference?

.    Line 102. Change « a serial of des mutants » to « a set of des mutants ». Mention here the meaning of des (Not defined before except in the title).

.    Line 111. « Arrows indicate [...] position ». Nothing related on the figure and no mention in the main text.

.    Line 128. Change « premature on » to « premature stop of »

.    Line 129. Change « no publications » to « no publication »

.    Line 161. « were clustered in a major clade ». A conclusion on this observation is missing.

.    Line 174. Change « nuclear » to « nucleus ».

.    Line 180. Change « DES » to « DES4 » (in italic).

            Phylogenetic tree of DES4. The alignment used to get this tree was done using full sequence or partial sequence?

.    Lines 182-183. The caption of the 2 pictures  of ISH seems to be in the wrong way.

.    Line 191. Change « to alkylates » to « to alkylate ».

.    Line 191Lines 201-202. «Given the [...] and DES4,». No clear evidence about this from the results. Rewrite this sentence.

.    Line 210. « We performed qRT-PCR experiments ». Where are RNA samples coming from? Which organs? Which stages? Have to be mentioned in the M&M.

.    Lines 217-220. So what? Is the divergent expression behaviour between cyclin genes coherent with DNA damage hypothesis?

.    Line 224. « elevated levels of Ku70 and Ku80 ». The differential expression between mutant and WT is very low. Difficult to get strong conclusion only based on this result.

.    Lines 226-232. There is no mention of RAD17 expression pattern in this part. Why?

Author Response

Response to reviewer 2

There is no clear evidence from the data that DES4 is involved in DNA repairing.

Response: since des4 is hypersensitive to the DNA damages induced by MMS or zeocine, we believe the gene should be related to DNA repairing either directly or indirectly. We understand that DES4 may not directly repair DNA damages, therefore we emphasized more about its “involvement” in this event, instead of stating “DES4 repairs DNA”.

Moreover DES4 can’t be considered as a spikelet number determination gene stricto sensu but rather a gene that may affect both vegetative and reproductive meristem functioning. In this sense, it would have been important to get a histological description of the meristems in des4 mutant in order to evidence a putative mis-organisation as reported in A. thaliana, as well as phenotypic analysis of spikelet, culm and root development in des4 mutant.

Response: similar to the issue above, we are talking about the “involvement”, instead of the “only and specific role”, of DES4 in spikelet number determination, which is the starting point and major focus of this project, though des4 displayed pleiotropical phenotypes. We totally agree that a histological analysis would help us to understand how the des4 meristem mis-organized. Unfortunately, it may take several months for us to have the tissues ready for sectioning. On the other hand, as the final consequence of defected meristem development, the obviously reduced spikelet numbers fully supported our major point that “DES3 is involved in spikelet number determination”, which has already made this paper a complete story. We are sorry that we can’t make it up this time, but will definitely finish it in our following-up work.

Panicle images from complementation assay plants and panicle branching mutants (Fig. 2 and Fig. S3) were shown but it would be nice to get scoring (in supplementary material?) as for the des4 mutant panicles (Fig. 1).

Response: data is provided as supplemental figure.

The results and discussion section related to the qRT-PCR data of marker genes is too speculative and in some cases the differential of expression between des4 and the WT is too small to be able to get clear conclusion. The authors should be more cautious about their conclusions.

Response: we rewrote the parts to avoid overstatement of our results.

Line 20. Change “premature of » by « premature stop of »;.    Line 21. Change “The des4 exhibited” by ‘The des4 mutant exhibited” Line 24. Change « nuclear » by « nucleus ». Change « serial » by « set »; Lines 25-27. This sentence is too long not clear.; Line 32. Oryza sativa in italic; Line 37. Delete « (also called inflorescence) ». 
 Panicle is one of the inflorescence types..    Line 48. Define SM (not mention before). Line 50. Change « progress has been achieved » to « progress achieved »; Line 55. Change « serial » to « set »; Line 58. Change « involve » to « are involved ».

Response: Corrected.

Lines 64-65. Which years ?

Response: From year 2014 to 2018.

Line 79. DES4 in italic; Line 90. Which Actin gene (LOC or RAPDB locus name) was used as qRT-PCR reference?

Response: We provided the actin gene ID (LOC_Os03g61970) here, which was actually also provided in the legend of figure 5 in our 1st submission.

Line 102. Change « a serial of des mutants » to « a set of des mutants ». Mention here the meaning of des (Not defined before except in the title). Line 111. « Arrows indicate [...] position ». Nothing related on the figure and no mention in the main text..    Line 128. Change « premature on » to « premature stop of »;.    Line 129. Change « no publications » to « no publication ».  Line 161. « were clustered in a major clade ». A conclusion on this observation is missing. Line 174. Change « nuclear » to « nucleus ». Line 180. Change « DES » to « DES4 » (in italic).

Response: Corrected.

Phylogenetic tree of DES4. The alignment used to get this tree was done using full sequence or partial sequence?

Response: We used full sequence for this assay.

Lines 182-183. The caption of the 2 pictures  of ISH seems to be in the wrong way.

Response: We double checked the result, it should be right.

Line 191. Change « to alkylates » to « to alkylate »..    Line 191Lines 201-202. «Given the [...] and DES4,». No clear evidence about this from the results. Rewrite this sentence.

Response: we rewrote this part.

Line 210. « We performed qRT-PCR experiments ». Where are RNA samples coming from? Which organs? Which stages? Have to be mentioned in the M&M.

Response: We used 14-day-old seedling samples for the qRT-PCR, details are provided in the result.

Lines 217-220. So what? Is the divergent expression behaviour between cyclin genes coherent with DNA damage hypothesis? Line 224. « elevated levels of Ku70 and Ku80 ». The differential expression between mutant and WT is very low. Difficult to get strong conclusion only based on this result.

Response: DNA damage gives rise to altered cell cycle progression and transcription of cell cycling genes, but not vice versa. For ku70 and 80, we detected statistically significant differences in expression, albeit the level change is not very high. We totally agree with you that the qRT-PCR evidence is weak and indirect to support the involvement of these genes or pathways in DES4 function or DNA repairing. In recognition of this, we carefully used vocabularies like “potential”, “suggest”, “imply” to interpret out data. However, using these data, we attempted to provide readers the hints that they are more likely to be involved, considering that at least they were altered by des4 in transcription. We included some sentences in this part to avoid overstatement.

Lines 226-232. There is no mention of RAD17 expression pattern in this part. Why?

Response: Descriptions regarding RAD17 are included. It is a motivation and warm encouragement for the authors to receive such detailed and suggestive comments on our paper. We sincerely appreciate your efforts.

Round 2

Reviewer 2 Report

Manuscript ID: genes-445709

Title: Decreased spikelets 4 encoding a novel tetratricopeptide repeat domain-containing protein is involved in DNA repairing and spikelet number determination in rice

Authors: Shen Ni, Zongzhu Li, Jiancheng Ying, Jian Zhang *, Hongqi Chen *

Submitted to section: Plant Genetics and Genomics.

Revised form

The authors addressed most of the comments/questions from the previous version. Two remaining remarks:

- Fig3: in my opinion, there is still an inversion between panel C and panel D according the legend (panicle meristems vs tiller).

- as qRT-PCRs were performed on 14-day old seedling, the lower expression of LAX1, LAX2 and MOC1 genes is more related to vegetative stage (tillering) than panicle and spikelet development in this experiment. 

Author Response

- Fig3: in my opinion, there is still an inversion between panel C and panel D according the legend (panicle meristems vs tiller).

Response: We checked the original files and found you are right. We are terribly sorry for this mistake. Thank you for your insistence.

- as qRT-PCRs were performed on 14-day old seedling, the lower expression of LAX1, LAX2 and MOC1 genes is more related to vegetative stage (tillering) than panicle and spikelet development in this experiment. 

Response: LAX genes are panicle-specific, we actually did the qRT-PCR for this three gene using young panicle cDNAs. We made it clear in the text. Thank you so much for your efforts.

Genes EISSN 2073-4425 Published by MDPI AG, Basel, Switzerland RSS E-Mail Table of Contents Alert
Back to Top